# Postnatal Growth Faltering: Growth and Height Improvement at Two Years in Children with Very Low Birth Weight between 2002–2017

**DOI:** 10.3390/children9121800

**Published:** 2022-11-23

**Authors:** Lara González-García, Laura Mantecón-Fernández, Marta Suárez-Rodríguez, Rosa Arias-Llorente, Sonia Lareu-Vidal, Aleida Ibáñez-Fernández, María Caunedo-Jiménez, Clara González-López, Eva Fernández-Morán, Belén Fernández-Colomer, Gonzalo Solís-Sánchez

**Affiliations:** 1Pediatrics Department, Hospital Álvarez-Buylla, 33600 Mieres, Spain; 2Pediatrics Department, Hospital Universitario Central de Asturias, 33011 Oviedo, Spain; 3Instituto Investigación Sanitaria Principado de Asturias, ISPA, 33011 Oviedo, Spain; 4Medical Department, University of Oviedo, 33003 Oviedo, Spain

**Keywords:** infant premature, very low birth weight, extrauterine growth restriction

## Abstract

The prevalence of postnatal growth faltering (PGF) in preterm infants with very low birth weight (VLBW) (<1500 g) is a universal problem. Growth improvement is expected as neonatal care is optimized. Objectives: To determine if there has been a decrease in the prevalence of PGF and an improvement in height at 2 years in appropriate for gestational age VLBW children in the last two decades. Methods: Clinical descriptive retrospective analysis of neonatal somatometry at birth and at two-year corrected age in VLBW preterm infants. Small for gestational age were excluded. Two cohorts (2002–2006, *n* = 112; and 2013–2017, *n* = 92) were compared. Results. In the second five-year period, a decrease in prevalence of PGF was observed (36.6% vs. 22.8%, *p* = 0.033), an increase in growth rate in the first 28 days (5.22 (4.35–6.09) g/kg/day vs. 11.38 (10.61–12.15) g/kg/day, *p* < 0.0001) and an increase in height standard deviation (SD) at 2 years (−1.12 (−1.35–−0.91) vs. −0.74 (−0.99–−0.49) *p* = 0.023). Probability of short stature at 2 years was directly related to daily weight gain in the first 28 days. Conclusions: when comparing two five-year periods in the last two decades, growth in VLBW preterm infants has improved, both during neonatal period and at two years of age.

## 1. Introduction

In recent decades, advances in neonatology and obstetrics have led to both increased survival and reduced morbidity in very low birth weight (VLBW) infants (<1500 g) [1,2,3,4,5]. However, although postnatal growth faltering (PGF) is still a universal health problem [6,7,8,9], it has also experienced significant improvement in recent decades [10].

PGF is defined as insufficient postnatal growth that occurs after preterm birth. There is no consensus in how to define PGF. It can be defined in two ways: transversal (static) or longitudinal (dynamic). Static PGF is usually defined at discharge or at 36/40 weeks postmenstrual age, including those patients weighing less than the 3rd or 10th percentile [11,12,13,14]. Dynamic PGF is defined as loss of 1 or 2 SD (standard deviation) between birth and a given time point (at discharge, 36 or 40 weeks usually). The lack of a reliable and agreed definition for PGF may lead to some children being misdiagnosed as PGF and being overfed, increasing their risk of obesity or cardiovascular disease [15]. In this study, we use the p10 for static PGF and the loss of plus −1 SD for dynamic PGF. PGF is considered an indicator of quality in neonatal units [16,17]. The etiology of PGF is multifactorial. It can be explained by inadequate nutrition, food intolerance, neonatal morbidity, and the rupture of the maternal-placental-fetal unit [7,18,19,20]. At discharge, it is estimated that around 50% of VLBW children will present static PGF using the INTERGROWTH-21st references [21]. If we exclude small for gestational age (SGA) children (weight < 10th percentile at birth), this prevalence ranges around 24–42% [21,22,23].

Traditionally, intrauterine growth references have been used as a reference for the postnatal growth of the premature [24]. However, in recent years the INTERGROWTH-21st growth references have been published, which, in addition to providing data for the assessment of neonatal somatometry, provide prospectively collected references on the neonatal longitudinal growth of premature infants with minimal morbidity and based on the latest recommendations [8,25,26].

The objective of our study was to assess whether there has been an improvement in height at 2 years of age in VLBW children when comparing two periods of time in recent decades in a cohort of Spanish VLBW children, and whether there was a decrease in the prevalence of PGF. SGA infants present growth impairment that does not have a postnatal origin in its entirety, but rather is the continuity of a process that began during fetal life [27]. For this reason, SGA children were excluded from the study.

## 2. Methods

A clinical descriptive retrospective study was designed. Two cohorts of patients that included premature infants (gestational age < 37 weeks) with birth weight less than 1500 g were separated in time into two five-years periods: 2002–2006 (quinquennium 1) and 2013–2017 (quinquennium 2).

They were all born in the Neonatology Unit of the Central University of Asturias (HUCA, Oviedo, Spain) and they were all included in a perinatal morbidity and two-year follow-up database (SEN 1500) after informed consent was given by their parents or legal guardian [28]. Our hospital is a tertiary level hospital that is reference for a population of one million inhabitants with about 5000 deliveries per year. Data at birth and 2 years were collected prospectively.

Exclusion criteria were: birth weight below the 10th percentile (INTERGROWTH-21st references), gestational age < 24 weeks, death, major congenital malformations, chromosomal abnormalities, congenital embryopathies, or the lack of outpatient follow-up until 24 ± 6 months of corrected age (follow-up is indicated at discharge in all patients). Static PGF was defined if the weight at discharge was less than the 10th percentile (INTERGROWTH-21st references) and dynamic PGF if there was a decrease of more than 1 SD in the weight (INTERGROWTH-21st references). The rate of weight gain in the first 28 days (g/kg/day) was calculated using the formula: (Weight (g) 28 days − Weight (g) birth) ÷ 28 ÷ Birth weight (kg).

At 2 years of corrected age, SD for weight, height, head circumference and BMI were calculated using the WHO charts [29]. BMI was calculated with the formula: (kg/m^2^). Short stature was defined if the height was less than –2 DS, malnutrition if the BMI was less than −2 SD and microcephaly if the head circumference was less than −2 SD.

### 2.1. Statistical Analysis

Data were analyzed using the IBM SPSS statistical software, version V22.0 (IBM^®^). Qualitative variables are shown as absolute number and percentage; the quantitative ones as mean and SD or as median and interquartile range. “Chi-square test” or “Fisher’s exact test” were used for the comparison of qualitative variables and the “Student’s *t*” test or the “Mann-Whitney U” test for quantitative variables. ROC curves were used to estimate the predictive power of short stature at 2 years based on the lowest daily weight gain in the first 28 days.

The significance level adopted was 5%.

### 2.2. Ethics

The study was carried out in accordance with good clinical practice and current legal regulations. The parents or legal guardians signed the informed consent before their inclusion in perinatal morbidity and two-year follow-up database (SEN 1500). The procedures performed in this study are part of the routine care of VLBW children. The data was managed ensuring the confidentiality of the data as stated in the Biomedical Research Law of 2007 (Law 14/2007) (www.boe.es/eli/es/l/2007/07/03/14 (accessed on 1 October 2022)). and the Law of Protection of Personal Data of 2018 (Law 3/2018) (www.boe.es/eli/es/lo/2018/12/05/3/con (accessed on 1 October 2022)). The study was approved by the Research Ethics Committee of the Principality of Asturias (No. 2020.314).

## 3. Results

A total of 237 VLBW children were enrolled in the first five-year period and 206 VLBW children in the second five-year period. In all, 112 and 92 appropriate for gestational age (AGA) VLBW children, respectively, met the inclusion criteria and completed the follow-up to 2 years. Figure 1 shows the flowchart with the patients included. A decrease in mortality was observed from 17% (*n* = 39) in quinquennium 1 to 10% (*n* = 20) in quinquennium 2 (*p* = 0.037), thus increasing survival from 83% to 90%.

Mean weight at birth was 1181 ± 238 g in quinquenium 1 and 1205 ± 211 in quinquennium 2 (differences were not significant). The characteristics of the population studied, as well as neonatal morbidity, are described in Table 1. In five-year period 2, a significant decrease was observed in intubation during resuscitation and in invasive ventilation, as well as an increase in the use of non-invasive ventilation. In addition, parenteral use at 28 days, necrotizing enterocolitis, periventricular leukomalacia and total length of stay decreased significantly. Table 2 shows the somatometric evolution during the neonatal period in both five-year periods, observing a higher weight SD at 28 days, at 36 weeks and at discharge in the second five-year period.

### 3.1. Decrease in the Prevalence of Extrauterine Growth Restriction

A statistically significant decrease was observed in the prevalence of PGF (static and dynamic) for weight and length, without changes in head circumference (see Table 3 and Figure 2). Weight gain in the first 28 days went from 5.2 (4.3–6.0) g/kg/day in quinquennium 1 to 11.3 (10.6–12.1) g/kg/day in quinquennium 2 (*p* < 0.0001).

### 3.2. Height Improvement at 2 Years Comparing the 2002–2006 Quinquennium vs. 2013–2017 Quinquennium

At 2 years, it was observed that the prevalence of short stature (<−2 SD) decreased from 20.5% in quinquennium 1 to 15.2% in quinquennium 2 (*p* = 0.32). Neither in the first nor in the second period studied were differences in the prevalence of height <−3 SD (4.5% vs. 4.4%, *p* = 1.00). Moreover, in quinquennium 2 there was both, a significant increase in the mean height expressed as SD (−1.12(−1.35–−0.91) vs. −0.74(−0.99–−0.49), *p* = 0.023). See Table 4 and Figure 3.

The probability of short stature (−2 SD) at 2 years was directly related to the daily weight gain in the first 28 days, as observed in the ROC curve: AUC = 0.606 (95% CI 0.503–0.708, *p* = 0.045). See Figure 4.

Analyzing weight, length and head circumference, SD between birth, discharge and 2 years, it is observed in both five-year periods that height is the most affected parameter at 2 years, and it is less likely to achieve an adequate catch up (Table 4 and Figure 5). However, the second five-year period shows a smaller drop in SD between birth and discharge and a better prognosis for height at 2 years. The mean height at 2 years in the second five-year period was 0.38 SD higher. When differentiating by gender, we observed that no differences were seen in the first five-year period (boys −1.12 ± 1.14 vs. girls −1.11 ±1.12, *p* > 0.05), nonetheless, there were differences in the second five-year period (boys −1.07 ± 1.24 vs. girls −0.50 ± 1.14, *p* = 0.024), girls presenting a height 0.61 SD higher.

### 3.3. Changes in the 2-Year Evolution of Patients with PGF Comparing 2002–2006 vs. 2013–2017

In quinquennium 1, patients who experienced PGF were more likely to have malnutrition (17.1 vs. 0%, *p* = 0.08). It is also observed that these patients have a higher probability of height <−2 SD, but these differences are not statistically significant. See Table 5 and Figure 6.

## 4. Discussion

This study was carried out in AGA VLBW children and shows their growth improvement during the neonatal period and in the first two years of age by comparing two quinquenniums in the last two decades. It is likely to be related to the effect of advances in the care of these patients, and has an impact not only on mortality but also leads to an improvement in the growth trajectory that is maintained at 2 years. In addition, we observed a direct relationship between daily weight gain in the first 28 days and the risk of short stature at 2 years. The importance of the inclusion criteria that we used when evaluating the growth of premature infants, should be considered. This study analyzes the growth of AGA VLBW children, which cannot be extrapolated when we compare with the total population of VLBW (30% SGA in our series) or with very preterm infants, as reflected in the review by Hollanders et al. [30].

We obtained a significant improvement in survival when comparing the 2002–2006 and 2013–2017 cohorts. These findings were similar to those observed in other studies [1,2,3,4,5]. Survival in 2002–2006 quinquennium was 83%, slightly higher than that described by Su et al. (81.5%) in the same five-year period in VLBW children, who also showed an increase in survival in the following analyzed period [4]. In addition to improving survival, one of the goals of neonatology is to reduce morbidity and improve the prognosis of premature infants. We have observed a clear improvement in the analyzed prognostic parameter, growth in the first 2 years and the prevalence of PGF. This improvement was observed in both, dynamic and static PGF. There is no consensus on which criteria should be used to define PGF or at what time (at discharge vs. 36–40 weeks postmenstrual age), which makes comparison between studies difficult. To establish a consensus, to define the definition of PGF would facilitate the comparison of results. The cause of this change in prognosis may be related to the reduction (rates reduced approximately to half) in the rates of intubation, resuscitation and the use of invasive ventilation (and a significant increase in the use of non-invasive ventilation). There is also a decrease in the need for parenteral nutrition at 28 days of life, a decrease in the prevalence of early onset sepsis, necrotizing enterocolitis, periventricular leukomalacia and in the length of stay. Other authors have observed similar results, establishing a relationship between periventricular leukomalacia [31], the longest duration of admission [32] and growth difficulties. A decrease in the prevalence of late onset sepsis, usually related to catheter use, has not been achieved in the second quinquennium, a point at which we consider that it could be improved. Similar results have been reported in other series, as described by Stoll et al. with preterm infants born between 2005 and 2012 [33]. A lower daily weight gain was also observed in the first 4 weeks in the first quinquennium, a parameter related to growth restriction [22,34]. We observed that the lower daily weight gain in the first 4 weeks was related to the risks of short stature at 2 years. Similarly, Borregas et al. [35] found significant differences in mean height at 2 years in VLBW children born between 2002–2015, depending on whether or not they had adequate in-hospital growth velocity. The overall prevalence of short stature was 20.5% in the first quinquennium and 15.2% in the second one, which have been similar to the results described in other series. Lebrão et al. found a 33.5% of prevalence in short stature at one year of life [36] and Takayanagui et al. found 11.8% at 6 years [37], varying the prevalence of short stature depending on the year of life analyzed, due to the catch up process.

In our series, girls experienced an improvement in height at 2 years, that was significantly higher than boys in the second quinquennium. Other authors have also described a higher risk of postnatal growth restriction in males [38], as well as a worse overall prognosis in VLBW boys, observing a longer length of stay and higher mortality [39]. The increase in the height of girls by 0.61 SD in the second five-year period is especially striking. In the event that this 0.67 SD height is maintained in adulthood, it would be equivalent to 4 cm more in adulthood according to the WHO graphs.

We also observed that patients who experienced PGF in the first quinquennium had a significantly higher risk of malnutrition at 2 years than those who experienced PGF in the second quinquennium, which could be explained by a greater severity of PGF in the first quinquennium. Other authors, such as Takayanagui et al., observed a relationship between PGF and short stature and malnutrition at 6 years of age in VLBW children [37].

Among the strengths of our study, we analyzed changes in the postnatal growth restriction, that has entirely a postnatal origin. We also use the new INTERGROWTH-21st references, based on the real longitudinal growth of premature infants with minimal morbidity. Among the limitations of our study, we must mention that it deals with data collected from a single center and the relatively small sample we analyzed. Additionally, we did not consider the height of the parents, the caloric intake received during admission or the different protocols applied in the neonatal unit for the management of premature infants in such a long period of time. We also consider as a limitation that the differences in the mean gestational age at birth and in the birth weight SD score that are observed between both five-year periods could influence the results. On the other hand, other factors such as maternal socioeconomic status were not included in the study. In addition to having excluded the SGA children, our results cannot be extrapolated with other VLBW series.

## 5. Conclusions

Based on these results, we can conclude that both growth in the neonatal period and growth at 2 years have improved, when comparing two five-year periods in different decades. The prognosis is better for girls compared to boys.

## Figures and Tables

**Figure 1 children-09-01800-f001:**
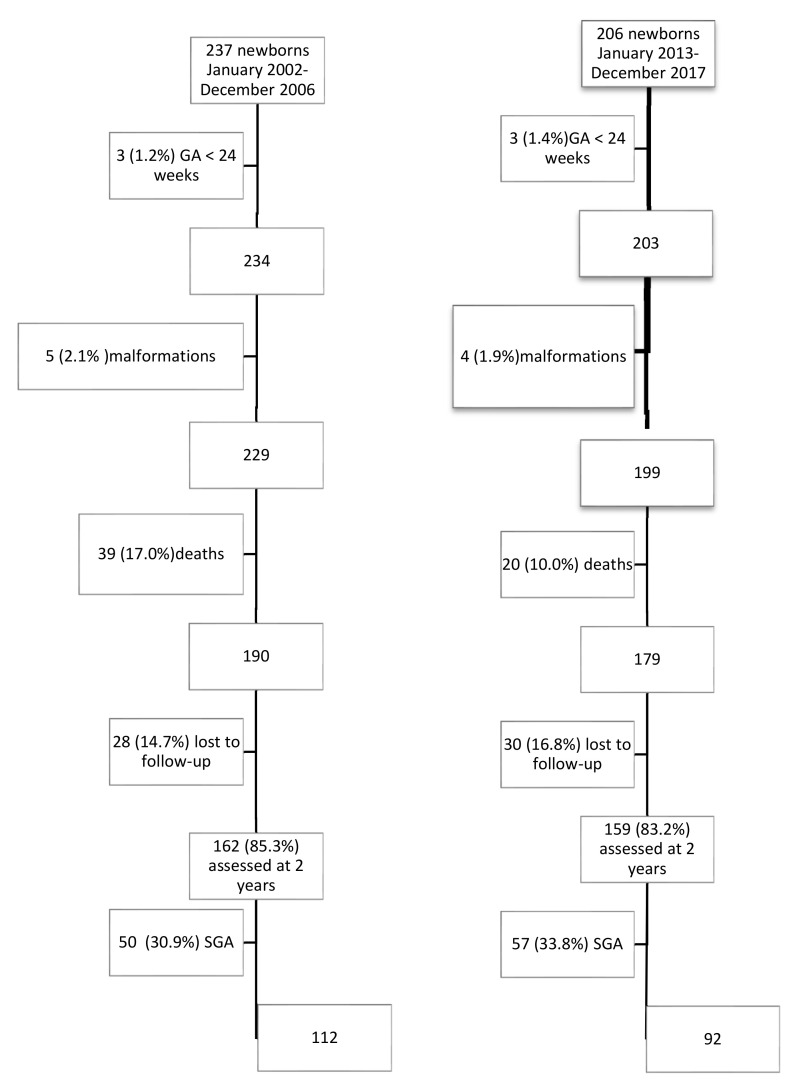
Flowchart of the sample according to inclusion criteria and study population. GA: gestational age. SGA: small for gestational age.

**Figure 2 children-09-01800-f002:**
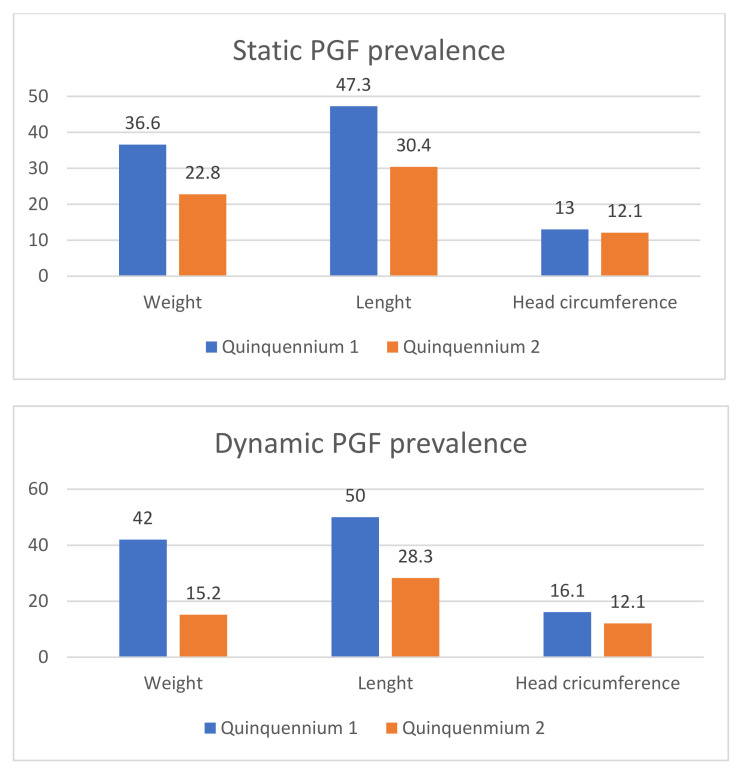
Changes in prevalence of static and dynamic postnatal growth faltering (PGF) (quinquennium 1: 2002–2006 vs. quinquennium 2: 2013–2017) in very low birth weight infants.

**Figure 3 children-09-01800-f003:**
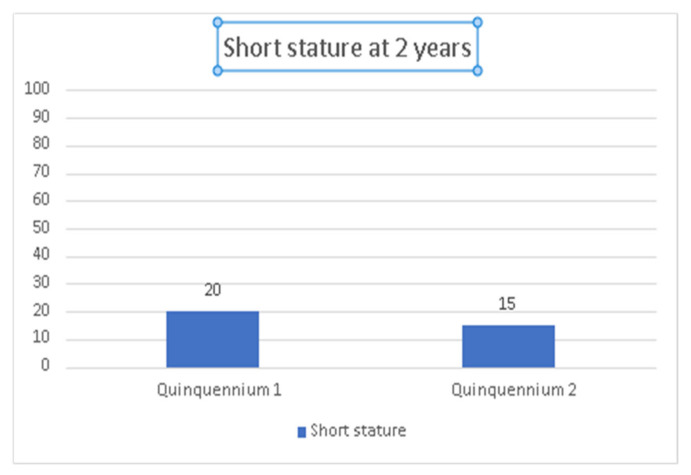
Prevalence of short stature (−2 SD) in very low birth weight infants (quinquennium 1: 2002–2006 vs. quinquennium 2: 2013–2017).

**Figure 4 children-09-01800-f004:**
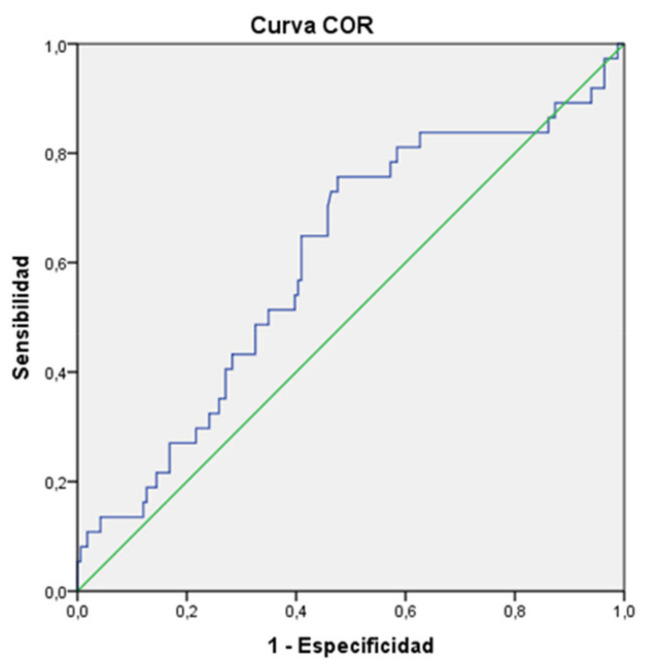
ROC curve: relationship between less weight gain in the first 28 days (g/kg/day) and risk of short stature (−2 SD).

**Figure 5 children-09-01800-f005:**
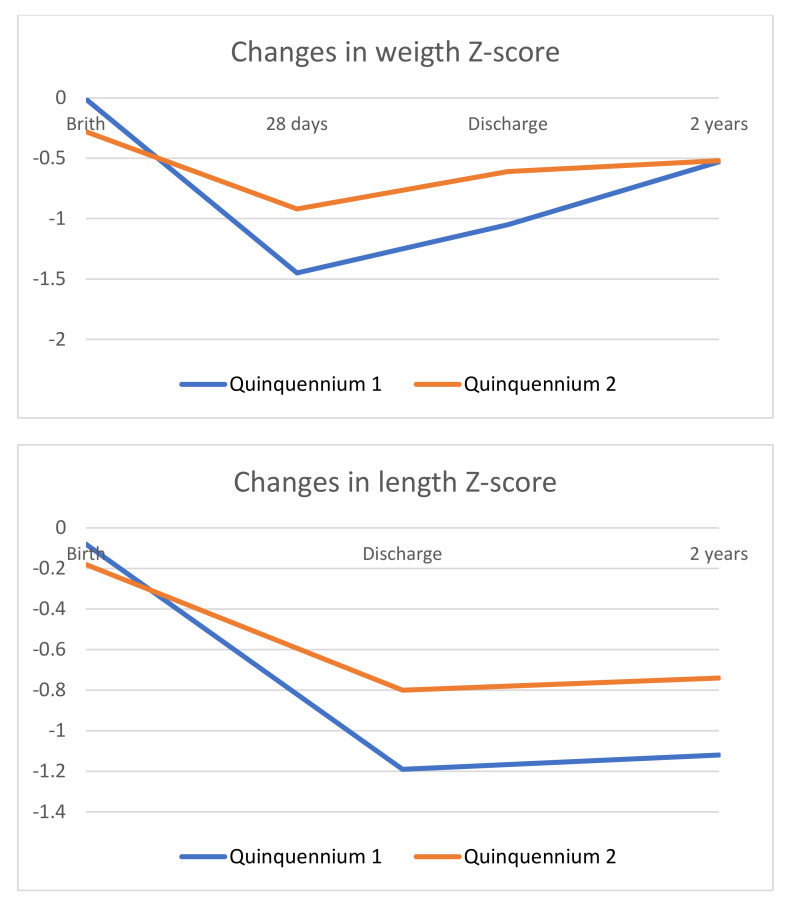
Evolution of standard deviation for weight, length, and head circumference at birth, hospital discharge, and 2 years (quinquennium 1: 2002–2006 vs. quinquennium 2: 2013–2017).

**Figure 6 children-09-01800-f006:**
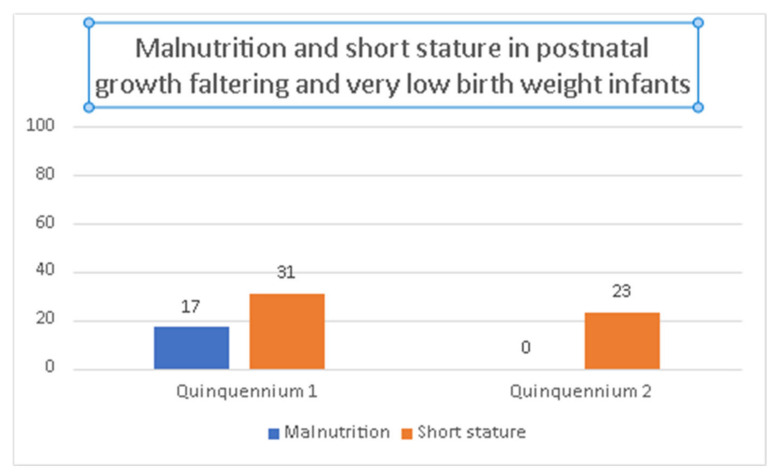
Prevalence of malnutrition and short stature at 2 years (quinquennium 1: 2002–2006 vs. quinquennium 2: 2013–2017) in postnatal growth faltering (PGF) and very low birth weight infants.

**Table 1 children-09-01800-t001:** Quinquennium 1 (2002–2006) vs. quinquennium 2 (2013–2017).

		Quinquennium 1	Quinquennium 2	*p*
	N	112	92	
Gestational age	At birth (weeks) ^2^	28.9 ± 2.0 (28.5–29.2)	29.5 ± 1.7 (29.1–29.9)	*** 0.017
	At discharge (weeks) ^2^	38.2 ± 2.8 (37.7–38.8)	37.7 ± 3.0 (37.1–38.3)	*** 0.19
Perinatal data	Male sex ^1^	51 (45.4)	39 (42.4)	* 0.65
	Antenatal corticosteroids ^1^	61 (55.0)	57 (62.0)	* 0.31
	Multiple gestation ^1^	38 (33.9)	34 (37.0)	* 0.65
	Caesarean section ^1^	71 (64.0)	60 (65.2)	* 0.85
	Apgar at 5 min < 5 ^1^	7 (6.3)	1 (1.1)	** 0.07
	Intubation in resuscitation ^1^	55 (49.5)	18 (19.6)	* <0.0001
Morbidity	RDS^1^	68 (60.7)	42 (45.7)	0.032
	Invasive mechanical ventilation ^1^	90 (80.4)	39 (42.4)	* <0.0001
	Non-invasive ventilation ^1^	69 (61.6)	89 (96.7)	* <0.0001
	Pneumothorax ^1^	5 (4.5)	2 (2.2)	** 0.46
	MV 28 days ^1^	10 (9)	2 (2.2)	* 0.04
	Early-onset sepsis ^1^	10 (8.9)	2 (2.2)	* 0.041
	Late-onset sepsis ^1^	40 (35.7)	36 (39.1)	* 0.61
	Parenteral nutrition at 28 days	19 (17)	4 (4.4)	0.005
	Necrotizing enterocolitis ^1^	8 (7.1)	0 (0)	** 0.009
	Patent ductus arteriosus ^1^	36 (32.1)	28 (30.4)	* 0.79
	Anemia (transfusion) ^1^	19 (39.6)	28(30.4)	* 0.27
	Acute kidney injury ^1^	1 (2.1)	5 (5.4)	** 0.66
	Hypotension (inotropics) ^1^	12 (10.7)	8 (8.7)	* 0.62
	Premature retinopathy ≥ stage 2 ^1^	9 (8.7)	5 (5.8)	* 0.44
	Bronchopulmonary dysplasia ^1^	30 (26.8)	16 (17.4)	* 0.11
	Periventricular leukomalacia ^1^	19 (17)	6 (6.5)	* 0.024
	HIV grade 3–4 ^1^	8 (7.1)	4 (4.3)	* 0.39
	Length of stay (days) ^3^	56 (30)	51 (24)	**** 0.004

Values are expressed as absolute number (%) ^1^ or mean ± standard deviation (95% IC) ^2^ or as median (interquartile range) ^3^ IVH: intraventricular hemorrhage, MV: mechanical ventilation. RDS: respiratory distress syndrome. Statistical analysis: * X2, ** Fisher’s exact test, *** *t*-student and **** U Mann-Withney.

**Table 2 children-09-01800-t002:** 2002–2006 vs. quinquennium 2: 2013–2017.

		Quinquennium 1	Quinquennium 2	*p*	OR (95% CI)
	N	112	92		
Somatometry at birth	Weight (g)^2^	1181 ± 238 (1136–1225)	1205 ± 211 (1162–1249)	0.44	
	Weight (SD) ^2^	−0.01 ± 0.69 (−0.14–0.11)	−0.28 ± 0.63 (−0.41–−0.14)	0.006	
	Length (cm) ^2^	37.9 ± 2.8 (37.4–38.5)	38.5 ± 2.8 (37.9–39.1)	0.18	
	Length (SD) ^2^	−0.08 ± 0.67 (−0.20–0.04)	−0.18 ± 0.74 (−0.33–−0.03)	0.30	
	Head circumference (cm) ^2^	26.2 ± 2.1 (25.8–26.6)	26.4 ± 2.0 (26.0–26.8)	0.39	
	HC (SD) ^2^	−0.21 ± 0.89 (−0.38–−0.05)	−0.42 ± 0.84 (−0.59–−0.24)	0.10	
	BMI (g/cm^2^) ^2^	8.0 ± 0.9 (7.8–8.2)	8.1 ± 0.7 (7.9–8.2)	0.64	
	BMI (SD) ^2^	−0.23 ± 0.88 (−0.39–−0.06)	−0.40 ± 0.83 (−0.57–−0.23)	0.15	
28 days	Weight ^2^	1369 ± 356 (1301–1436)	1595 ± 326 (1527–1663)	<0.0001	
	Weight (SD) ^2^	−1.45 ± 0.87 (−1.62–−1.29)	−0.92 ± 0.85 (−1.1–−0.74)	<0.0001	
	Weight SD < p10 ^2^	66 (59.5)	33 (35.9)	0.001	2.62 (1.48–4.63)
	Weigth gain velocity g/kg/day 28 days ^2^	5.2 ± 4.6 (4.3–6.0)	11.3 ± 3.7 (10.6–12.1)	<0.0001	
36 weeks	Weight ^2^	1907 ± 315 (1845–1968)	2104 ± 266 (2046–2161)	<0.0001	
	Weight (SD) ^2^	−1.53 ± 1.20 (−1.77–−1.29)	−0.84 ± 0.84 (−1.03–−0.66)	<0.0001	
Somatometry at discharge	Weight (g) ^2^	2454 ± 248 (2407–2500)	2502 ± 427 (2413–2590)	0.34	
	Weight (SD) ^2^	−1.05 ± 1.33 (−1.30–0.80)	−0.61 ± 0.92 (−0.80–−0.42)	0.008	
	Length (cm) ^2^	45.9 ± 1.9 (45.5–46.3)	46.0 ± 2.7 (45.4–46.6)	0.71	
	Length (SD) ^2^	−1.19 ± 1.64 (−1.50–−0.88)	−0.80 ± 1.55 (−1.12–−0.47)	0.08	
	Head circumference (cm) ^2^	33.3 ± 1.1 (33.0–33.5)	33.2 ± 2.1 (32.7–33.6)	0.74	
	HC (SD) ^2^	−0.12 ± 1.49 (−0.40–0.15)	0.06 ± 1.74 (−0.29–0.42)	0.40	
	BMI (g/cm^2^) ^2^	11.5 ± 0.9 (11.4–11.7)	11.7 ± 1.5 (11.4–12.1)	0.25	
	BMI (SD) ^2^	−0.72 ± 0.83 (−0.88–−0.57)	−0.43 ± 1.01 (−0.64–−0.22)	0.027	

Values are expressed as absolute number (%) or as mean ± standard deviation (95% IC) ^2^. HC: head circumference. Statistical analysis: X2 or *t*-student.

**Table 3 children-09-01800-t003:** Evolution of the prevalence of static postnatal growth faltering (PGF) (<p10 at discharge) and dynamic PGF (decrease in weight > 1 DS) in AGA VLBW children (quinquennium 1: 2002–2006 vs. quinquennium 2: 2013–2017).

		Quinquennium 1	Quinquennium 2	*p*	OR (95% CI)
	N	112	92		
PGF weight	<p10	41 (36.6)	21 (22.8)	0.033	1.95 (1.05–3.63)
	>−1 DS	47 (42.0)	14 (15.2)	<0.001	4.02 (2.03–7.96)
PGF length	<p10	53 (47.3)	28 (30.4)	0.014	2.05 (1.15–3.66)
	>−1 DS	56 (50.0)	26 (28.3)	0.002	2.53 (1.41–4.56)
PGF HC	<p10	16 (14.3)	12 (13.0)	0.79	
	>−1 DS	18 (16.1)	11 (12.1)	0.50	

Values are expressed as absolute number (%). PGF: postnatal growth faltering. HC: Head circumference.

**Table 4 children-09-01800-t004:** Change in neonatal somatometry (quinquennium 1: 2002–2006 vs. quinquennium 2: 2013–2017).

		Quinquennium 1	Quinquennium 2	*p*
Assessment 2 years	N	112	92	
	Chronological age (months) ^2^	26.33 ± 1.78 (26.00–26.67)	26.85 ± 2.94 (26.33–27.36)	*** 0.09
	Corrected age (months) ^2^	23.76 ± 1.74 (23.43–24.08)	24.40 ± 2.41 (23.9–24.9)	*** 0.034
Somatometry 2 years	Weight (kg) ^2^	11.23 ± 1.55 (10.94–11.53)	11.46 ± 1.50 (11.15–11.77)	*** 0.30
	Weight (SD) ^2^	−0.53 ± 1.17 (−0.75–−0.31)	−0.52 ± 1.09 (−0.74–−0.29)	*** 0.96
	Height (cm) ^2^	85.2 ± 3.8 (84.5–86.0)	85.4 ± 4.0 (84.5–86.2)	*** 0.80
	Height (SD) ^2^	−1.12 ± 1.11 (−1.35–−0.91)	−0.74 ± 1.21 (−0.99–−0.49)	*** 0.023
	Short stature ^1^	23 (20.5)	14 (15.2)	* 0.32
	HC (cm) ^2^	48.50 ± 1.77 (48.17–48.84)	48.32 ± 2.01 (47.90–48.73)	* 0.48
	Microcephaly ^1^	3 (2.7)	6 (6.7)	** 0.19
	HC (SD) ^2^	0.57 ± 1.16 (0.35–0.79)	0.26 ± 1.52 (−0.05–0.58)	* 0.09
	BMI ^2^	15.62 ± 1.37 (15.36–15.87)	15.73 ± 1.44 (15.43–16.03)	* 0.55
	BMI (SD) ^2^	−0.22 ± 1.09 (−0.43–−0.02)	0.13 ± 1.31 (−0.14–0.40)	* 0.034
	Malnutrition ^1^	8 (9)	2 (2.2)	** 0.11

Values are expressed as absolute number (%) ^1^ or mean ± standard deviation (95% IC) ^2^. HD: head circumference. Statistical analysis: * X2, ** Fisher’s exact test and *** *t*-student.

**Table 5 children-09-01800-t005:** Somatometry at 2 years in postnatal growth faltering and appropriate for gestational age very low birth weight infants (quinquennium 1: 2002–2006 vs. quinquennium 2: 2013–2017).

	Quinquennium 1	Quinquennium 2	*p*
Short stature	13 (31.7)	5 (23.8)	0.51
Malnutrition	7 (17.1)	0 (0)	0.08
Microcephaly	3 (7.3)	2 (9.5)	1

Values are expressed as absolute number (%).

## Data Availability

The data is on file with LGG and is available to potential researchers through the corresponding author.

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
