# Peer review of "Postnatal Growth Faltering: Growth and Height Improvement at Two Years in Children with Very Low Birth Weight between 2002–2017"

_children, 2022, doi:10.3390/children9121800_

Round 1

Reviewer 1 Report

This is an interesting paper describing the degree of EUGR after VLBW as well as studying the impact of EUGR on infant growth between 2 epochs.

I only have few comments:

1. Methods. Still there is no universal definition of EUGR. The authors should explain the rationale behind their definition.

2. Methods. The ethics paragraph is almost empty. Simply stating that the study was approved by a review board makes no sense. If I understand it correctly, all procedures were part of routine clinical care and no study interventions took place. How were parents informed about the study? Did they have to give written informed consent for, or were they offered the chance to object against, the use of their child's data for this study? 

3. Overall. Throughout the manuscript, the authors should emphasize that their findings pertain to VLBW infants, which are in many respects different from very preterm infants. As a general rule, in VLBW populations there is an overrepresentation of SGA infants (which were not part of the studied sample!).

4. Discussion. In view of my previous comment, the authors should add a paragraph to the Discussion describing the impact of the choice of definition on subsequent growth (see also: Hollanders JJ, et al. Pediatr Res 2017, 82:317-23).

5. Discussion. The limitation section should be expanded; eg, there is no report of the relatively small size.

6. Discussion. It would be helpful if the authors added a paragraph to the Discussion describing the wealth of definitions for EUGR, complicating comparison across studies, and emphasizing the need for a consensus definition.

Author Response

Responses to reviewer 1

Thank you very much for your kind words. Without a doubt, the article will improve a lot with all the suggested contributions.

I will now respond to the suggestions received:

  1. Methods. Still there is no universal definition of EUGR. The authors should explain the rationale behind their definition.

There is no universal definition for EUGR. Static EUGR is usually defined at discharge, or at 36 or 40 weeks, including those patients weighing less than the 3rd or 10th percentile. Dynamic EUGR is defined as loss of 1 or 2 SD between birth and a given time point. (at discharge, 36 or 40 weeks usually). In this study we use the p10 for static EUGR and the loss of plus -1 SD for dynamic EUGR

            Ordóñez-Díaz MD, Pérez-Navero JL, Flores-Rojas K, Olza-Meneses J, Muñoz-Villanueva MC, Aguilera-García CM, et al. Prematurity With Extrauterine Growth Restriction Increases the Risk of Higher Levels of Glucose, Low-Grade of Inflammation and Hypertension in Prepubertal Children. Front Pediatr. 21 de abril de 2020;8:180.

            Tozzi MG, Moscuzza F, Michelucci A, Lorenzoni F, Cosini C, Ciantelli M, et al. ExtraUterine Growth Restriction (EUGR) in Preterm Infants: Growth Patterns, Nutrition, and Epigenetic Markers. A Pilot Study. Frontiers in Pediatrics [Internet]. 2018 [cita-do 27 de octubre de 2022];6. Disponible en: https://www.frontiersin.org/articles/10.3389/fped.2018.00408

            Maheshwari A, Bagga N, Panigrahay N. Extra-uterine Growth Restriction in Preterm Infants. Newborn. 31 de marzo de 2022;1(1):67-73.

            Kim YJ, Shin SH, Cho H, Shin SH, Kim SH, Song IG, et al. Extrauterine growth restriction in extremely preterm infants based on the Intergrowth-21st Project Preterm Postnatal Follow-up Study growth charts and the Fenton growth charts. Eur J Pediatr. 9 de septiembre de 2020;

  1. Methods. The ethics paragraph is almost empty. Simply stating that the study was approved by a review board makes no sense. If I understand it correctly, all procedures were part of routine clinical care and no study interventions took place. How were parents informed about the study? Did they have to give written informed consent for, or were they offered the chance to object against, the use of their child's data for this study? 

Thank you very much for your comment. Certainly, we had not detailed an aspect that is relevant. Now we have proceeded to clarify them.

The study was carried out in accordance with good clinical practice and current legal regulations. The parents or legal guardians signed the informed consent before their in-clusion in perinatal morbidity and two-year follow-up database (SEN 1500). The proce-dures performed in this study are part of the routine care of VLBW children, therefore waiver of informed consent for participation in this study was granted. The data was managed ensuring the confidentiality of the data as stated in the Biomedical Research Law of 2007 (Law 14/2007) (www.boe.es/eli/es/l/2007/07/03/14) and the Law of Protection of Personal Data of 2018 (Law 3/2018) (www.boe.es/eli/es/lo/2018/12/05/3/con). The study was approved by the Research Ethics Committee of the Principality of Asturias (No. 2020.314).

  1. Overall. Throughout the manuscript, the authors should emphasize that their findings pertain to VLBW infants, which are in many respects different from very preterm infants. As a general rule, in VLBW populations there is an overrepresentation of SGA infants (which were not part of the studied sample!).

Thank you very much for your comment. We have emphasized in the text that the data refer to VLBW children. We believe that the manuscript is now better clarified.

  1. Discussion. In view of my previous comment, the authors should add a paragraph to the Discussion describing the impact of the choice of definition on subsequent growth (see also: Hollanders JJ, et al. Pediatr Res 2017, 82:317-23).

Thank you very much for the contribution and for the reference. We have included it in the discussion.

  1. Discussion. The limitation section should be expanded; eg, there is no report of the relatively small size.

Thank you very much for the comment. We have added this limitation to the discussion and expanded the limitations of the study.

  1. Discussion. It would be helpful if the authors added a paragraph to the Discussion describing the wealth of definitions for EUGR, complicating comparison across studies, and emphasizing the need for a consensus definition.

Thank you very much for your comment. The lack of consensus on the definition of EUGR led us to analyze both dynamic and static EUGR. Indeed, we consider it essential to establish a consensus to define EUGR. We have incorporated this idea into the discussion.

Reviewer 2 Report

It is not true that EUGR associated with long-term growth difficulties as noted in a recent paper by several neonatal experts who recently reviewed the literature regarding EUGR and founds it is not associated with long-term outcomes. They provided five or six reasons why EUGR is not a useful growth metric for preterm infants (‘Extrauterine growth restriction’ … are misnomers for preterm infants. doi:10.1038/s41372-020-0658-5) so it is unclear whether this submitted work provides value focused on EUGR.

The authors state that the rate of weight gain in the first 28 days (g/kg/day) was calculated using the formula: (Weight (g)28 days-Weight (g)birth)÷28÷Birth weight (kg). This is the “Early” calculation that gives inflated weight gain estimates since it divides by the denominator of birthweight, a small value. The numerical values of weight gain of 5 and 11 g/kg/day are not believable for this calculation method.

ROC curves use categorical outcomes. It is a good idea to use WHO charts and SD scores, however, no one (including the WHO) identify that -1 SD is a valid outcome of concern. Rather, the WHO uses <-2 SD as an outcome of short stature and <-3 SD as severe stunting.  What cut point was used to categorize weight gain as the “lowest daily gain” in the first 28 days? If the lowest values were used, what numbers of babies were in the small cells of the table?

The abstract needs to mention that SGA infants were excluded. Why were they excluded? Why not stratify by SGA and non-SGA since their growth patterns differ? It is questionable whether the Intergrowth data should be used to assign SGA since IUGR infants were excluded from their study and the American Academy of Pediatrics recommends that this data not be used for infants <36 weeks gestational age since the numbers <36 weeks were too small.

The paper switches back and forth between referring to z-scores and SDs; one term should be used consistently.

It is desirable that 3 significant digits are used for the p-values, as done, but the exact p-values should be used for non-significant results with 2 significant digits. The analysis emphasizes statistical significance over importance of the results, which is concerning for a multiple variable analysis such as this. Include 95% CI (not just p-values) for results. 

I am concerned with the conclusions being made from this paper that did not adjust for any confounding which reduces the internal validity of results and conclusions. There were important differences in the immaturity of the infants (gestational age and birthweight z-scores) and the care of the infants (respiratory management, parenteral nutrition, NEC, sepsis, brain injury, length of stay). Changes in practice and how preterm babies are cared for may have occurred over the two time periods that may be associated with long-term growth which were not accounted for in the analysis. Also, many baseline factors (maternal socio-economic status, gestational age at birth, etc) which could be confounders of the relationship in question were not included in any analysis. Therefore it is not valid to attribute the differences to growth, which was likely altered by the same factors that altered the other outcomes. The sentence “Neonatal growth in the first 28 days of life seems to be decisive” suggests causality, which is not appropriate especially since there were likely other factors that led to the growth differences that were not described in this study. Assuming causality is not appropriate since this is an observational study. 

The flow sheet should include percentages for losses to follow up for each reason.

Author Response

Responses to reviewer 2

Thank you very much for your kind words. Without a doubt, the article will improve a lot with all the suggested contributions.

I will now respond to the suggestions received:

  1. It is not true that EUGR associated with long-term growth difficulties as noted in a recent paper by several neonatal experts who recently reviewed the literature regarding EUGR and founds it is not associated with long-term outcomes. They provided five or six reasons why EUGR is not a useful growth metric for preterm infants (‘Extrauterine growth restriction’ … are misnomers for preterm infants. doi:10.1038/s41372-020-0658-5) so it is unclear whether this submitted work provides value focused on EUGR.

Thank you very much for your thoughtful comment. We have modified it in the text and added the reference. We have highlighted the importance of the lack of consensus in the definition of EUGR or the consequences of the erroneous diagnosis of EURG.

  1. The authors state that the rate of weight gain in the first 28 days (g/kg/day) was calculated using the formula: (Weight (g)28 days-Weight (g)birth)÷28÷Birth weight (kg). This is the “Early” calculation that gives inflated weight gain estimates since it divides by the denominator of birthweight, a small value. The numerical values of weight gain of 5 and 11 g/kg/day are not believable for this calculation method.

Thank you very much for your comment. There is no optimal way to define growth velocity as discussed in the review by Fenton et al (Fenton TR, Chan HT, Madhu A, et al. Preterm Infant Growth Velocity Calculations: A Systematic Review. Pediatrics. 2017;139(3):e20162045. doi:10.1542/peds.2016-2045). The fact that this formula may give exaggerated estimates of weight gain could lead to artifacts if we were to compare two different formulas. By using the same formula to analyze weight gain in the first 28 days in both five-year periods, we consider that it gives validity to the test when comparing results. The results obtained for weight gain (5 and 11 g/kg/day) are the real results obtained in our database and the ones that led us to consider this review of the subject, upon observing such a striking difference.

  1. ROC curves use categorical outcomes. It is a good idea to use WHO charts and SD scores, however, no one (including the WHO) identify that -1 SD is a valid outcome of concern. Rather, the WHO uses <-2 SD as an outcome of short stature and <-3 SD as severe stunting.  What cut point was used to categorize weight gain as the “lowest daily gain” in the first 28 days? If the lowest values were used, what numbers of babies were in the small cells of the table?

Thank you very much for such a wise comment.

As the reviewer points out, height less than -1 SD is not defined as a parameter that implies pathology. However, given that the population mean is the trend towards 0 SD, we consider that the significant difference in prevalence in height < -1 SD observed contributes to showing the improvement in height of the two cohorts analyzed.

We have added the difference in prevalence in height < -3 SD in the text (we did not observe significant differences in both periods).

We have corrected the expression in the text "lowest daily gain” in the first 28 days, since it is not adequate, we believe that the text is now much more precise. The ROC curve related the weight gain in the first 28 days with the diagnoses of short stature at 2 years. We obtained the area under the curve including all the values of growth rate in g/kg/day obtained in both five-year periods. Therefore, a cut-off point for growth rate was not used to make the ROC curve, otherwise all the values of the children analyzed were included.

  1. The abstract needs to mention that SGA infants were excluded. Why were they excluded? Why not stratify by SGA and non-SGA since their growth patterns differ? It is questionable whether the Intergrowth data should be used to assign SGA since IUGR infants were excluded from their study and the American Academy of Pediatrics recommends that this data not be used for infants <36 weeks gestational age since the numbers <36 weeks were too small.

Thank you very much for your kind comment. We have added the exclusion of SGA children in the abstract. They were excluded because SGA children include those children who have experienced IUGR, and these patients will present growth disturbance that originated prenatally and continues postnatally. The objective of this study was to compare postnatal growth up to 2 years in children with very low birth weight in two different five-year periods. Those children who were not IUGR can reach their full growth potential, and improvements in the care received will therefore be better reflected in these patients. Excluding SGA patients seems to us to focus attention on growth impairment that has a postnatal origin and therefore can be affected by neonatal praxis. Constitutionally small (non-IUGR SGA) children were excluded using the 10th percentile cut-off as a study limitation.

As he rightly comments, the intergrowth references were made excluding those children with evidence of IUGR. For this reason, we believe that they can better estimate the 10th percentile in the population, especially in the most premature children in whom a higher prevalence of IUGR is common. Although intergrowth was carried out analyzing a small number of premature infants, we believe that the precision of the method with which the references were obtained gives it validity.

  • The paper switches back and forth between referring to z-scores and SDs; one term should be used consistently.

Thank you very much for your comment. We have made the modification in the text, and we consider that the text has improved its precision.

  1. It is desirable that 3 significant digits are used for the p-values, as done, but the exact p-values should be used for non-significant results with 2 significant digits. The analysis emphasizes statistical significance over importance of the results, which is concerning for a multiple variable analysis such as this. Include 95% CI (not just p-values) for results. 

Thank you very much for such a good suggestion. We have corrected it and added it to the text.

  1. I am concerned with the conclusions being made from this paper that did not adjust for any confounding which reduces the internal validity of results and conclusions. There were important differences in the immaturity of the infants (gestational age and birthweight z-scores) and the care of the infants (respiratory management, parenteral nutrition, NEC, sepsis, brain injury, length of stay). Changes in practice and how preterm babies are cared for may have occurred over the two time periods that may be associated with long-term growth which were not accounted for in the analysis. Also, many baseline factors (maternal socio-economic status, gestational age at birth, etc) which could be confounders of the relationship in question were not included in any analysis. Therefore it is not valid to attribute the differences to growth, which was likely altered by the same factors that altered the other outcomes. The sentence “Neonatal growth in the first 28 days of life seems to be decisive” suggests causality, which is not appropriate especially since there were likely other factors that led to the growth differences that were not described in this study. Assuming causality is not appropriate since this is an observational study. 

Thank you very much for your interesting comment. Certainly, we must point out all the limitations that have been in the study, and so we have modified it in the manuscript. We have also corrected the causal relationship that we described in the text, since, as you say, it is not correct.

  1. The flow sheet should include percentages for losses to follow up for each reason.

Thanks a lot for the suggestion. we have included it in the flow chart.

Round 2

Reviewer 2 Report

The response was pleasant, the abstract conclusion greatly improved, some positive changes were made to the manuscript but it does not make sense to use a loss of >1 SD as a cut off as universally accepted cut points such as a loss >2 SD is pretty universally accepted and can be referenced by +/- 2 SD for the healthist of preterm infants (Rochow 2016 PMID: 26859363) and Goldberg et al 2018 recommended this cut point (although this is yet to be validated. This should be changed.

5 and 11 g/kg/day are not at all believable for the reported calculation formula. Please check your formula and calculation, then provide for the reviewers and the editor the mean values and number of days for both groups to show that the numbers can be believed. There is no need to use 2 decimal points for the growth rate estimates as that degree of precision is not needed or valid.

Given the advice to not use the phrase EUGR, perhaps the authors would agree to use instead “postnatal growth faltering”?

Author Response

  • The response was pleasant, the abstract conclusion greatly improved, some positive changes were made to the manuscript but it does not make sense to use a loss of >1 SD as a cut off as universally accepted cut points such as a loss >2 SD is pretty universally accepted and can be referenced by +/- 2 SD for the healthist of preterm infants (Rochow 2016 PMID: 26859363) and Goldberg et al 2018 recommended this cut point (although this is yet to be validated. This should be changed.

Thank you very much for your kind comment. We have highlighted the new modifications in green. We have already corrected the use of the cut-off point for height at -1 SD, focusing on -2 SD as reported in the literature.

  • 5 and 11 g/kg/day are not at all believable for the reported calculation formula. Please check your formula and calculation, then provide for the reviewers and the editor the mean values and number of days for both groups to show that the numbers can be believed. There is no need to use 2 decimal points for the growth rate estimates as that degree of precision is not needed or valid.

Thank you very much for your comment. Of course, we provide here the growth data in the first 28 days:

Quinquennium 1

Mean birth weight: 1181±238 g

Mean weight at 28 days: 1369±356 g

Average weight gain in the first 28 days: 6.6±6.0 g/day

Number of days analyzed: 28

Gain g/kg/day in first 28 days by birthweight:  5.2±4.6 g/kg/day

Quinquennium 2

Mean birth weight:  1205±211 g

Mean weight at 28 days: 1595±326 g

Average weight gain in the first 28 days: 13.9±5.3 g/day

Gain g/kg/day in first 28 days by birthweight: 11.3±3.7 g/kg/day

Number of days analyzed: 28

  • Given the advice to not use the phrase EUGR, perhaps the authors would agree to use instead “postnatal growth faltering”?

Thank you very much for your comment. We have replaced the term EUGR with postnatal growth faltering as suggested.
